# The Role of Food Insecurity and Dietary Diversity on Recovery from Wasting among Hospitalized Children Aged 6–23 Months in Sub-Saharan Africa and South Asia

**DOI:** 10.3390/nu14173481

**Published:** 2022-08-24

**Authors:** Adino Tesfahun Tsegaye, Patricia B. Pavlinac, Lynnth Turyagyenda, Abdoulaye H. Diallo, Blaise S. Gnoumou, Roseline M. Bamouni, Wieger P. Voskuijl, Meta van den Heuvel, Emmie Mbale, Christina L. Lancioni, Ezekiel Mupere, John Mukisa, Christopher Lwanga, Michael Atuhairwe, Mohammod J. Chisti, Tahmeed Ahmed, Abu S.M.S.B. Shahid, Ali F. Saleem, Zaubina Kazi, Benson O. Singa, Pholona Amam, Mary Masheti, James A. Berkley, Judd L. Walson, Kirkby D. Tickell

**Affiliations:** 1Department of Epidemiology, University of Washington, Seattle, WA 98195, USA; 2Departments of Global Health, University of Washington, Seattle, WA 98195, USA; 3Uganda-CWRU Research Collaboration, Kampala P.O. Box 663, Uganda; 4Department of Public Health, University Joseph Ki-Zerbo, Ouagadougou 03BP7021, Burkina Faso; 5Amsterdam UMC, University of Amsterdam, Amsterdam Centre for Global Child Health & Emma Children’s Hospital, Meibergdreef 9, 1105 AZ, Amsterdam, The Netherlands; 6Division of Paediatric Medicine, Hospital for Sick Children, Toronto, ON M5G 1X8, Canada; 7Department of Paediatrics and Child Health, Kamuzu University of Health Sciences, Private Bag 360, Chichiri, Blantyre 265, Malawi; 8Department of Pediatrics, Oregon Health and Science University, Portland, OR 97239, USA; 9Department of Paediatrics and Child Health, Makerere University College of Health Sciences, Kampala P.O. Box 7072, Uganda; 10Nutrition and Clinical Services Division, International Centre for Diarrhoeal Disease Research Bangladesh, Dhaka 1212, Bangladesh; 11Department of Pediatrics and Child Health, Aga Khan University, Karachi 74800, Pakistan; 12Kenya Medical Research Institute, Nairobi 54840, Kenya; 13KEMRI/Wellcome Trust Research Programme, Kilifi 80108, Kenya; 14The Childhood Acute Illness and Nutrition Network (CHAIN), Nairobi 184742, Kenya; 15Centre for Tropical Medicine and Global Health, Nuffield Department of Medicine, University of Oxford, Oxford OX3 7LG, UK

**Keywords:** wasting, hospitalization, food security, dietary diversity

## Abstract

Background: Current guidelines for the management of childhood wasting primarily focus on the provision of therapeutic foods and the treatment of medical complications. However, many children with wasting live in food-secure households, and multiple studies have demonstrated that the etiology of wasting is complex, including social, nutritional, and biological causes. We evaluated the contribution of household food insecurity, dietary diversity, and the consumption of specific food groups to the time to recovery from wasting after hospital discharge. Methods: We conducted a secondary analysis of the Childhood Acute Illness Network (CHAIN) cohort, a multicenter prospective study conducted in six low- or lower-middle-income countries. We included children aged 6–23 months with wasting (mid-upper arm circumference [MUAC] ≤ 12.5 cm) or kwashiorkor (bipedal edema) at the time of hospital discharge. The primary outcome was time to nutritional recovery, defined as a MUAC > 12.5 cm without edema. Using Cox proportional hazards models adjusted for age, sex, study site, HIV status, duration of hospitalization, enrollment MUAC, referral to a nutritional program, caregiver education, caregiver depression, the season of enrollment, residence, and household wealth status, we evaluated the role of reported food insecurity, dietary diversity, and specific food groups prior to hospitalization on time to recovery from wasting during the 6 months of posthospital discharge. Findings: Of 1286 included children, most participants (806, 63%) came from food-insecure households, including 170 (13%) with severe food insecurity, and 664 (52%) participants had insufficient dietary diversity. The median time to recovery was 96 days (18/100 child-months (95% CI: 17.0, 19.0)). Moderate (aHR 1.17 [0.96, 1.43]) and severe food insecurity (aHR 1.14 [0.88, 1.48]), and insufficient dietary diversity (aHR 1.07 [0.91, 1.25]) were not significantly associated with time to recovery. Children who had consumed legumes and nuts prior to diagnosis had a quicker recovery than those who did not (adjusted hazard ratio (aHR): 1.21 [1.01,1.44]). Consumption of dairy products (aHR 1.13 [0.96, 1.34], *p* = 0.14) and meat (aHR 1.11 [0.93, 1.33]), *p* = 0.23) were not statistically significantly associated with time to recovery. Consumption of fruits and vegetables (aHR 0.78 [0.65,0.94]) and breastfeeding (aHR 0.84 [0.71, 0.99]) before diagnosis were associated with longer time to recovery. Conclusion: Among wasted children discharged from hospital and managed in compliance with wasting guidelines, food insecurity and dietary diversity were not major determinants of recovery.

## 1. Background

Novel interventions to improve recovery from wasting are critically needed to reduce pediatric morbidity and mortality in resource-limited settings [1,2,3]. Community-based nutritional rehabilitation programs have been shown to be effective for managing childhood wasting without medical complications [4]. However, the nutritional recovery appears slower among children discharged from the hospital compared with children treated in the community, although the reasons for this slower rate of recovery are not clearly understood [5,6]. Multiple factors, including differences in the etiology of wasting among individual children, the presence of comorbidities, and socioeconomic vulnerabilities, all influence nutritional recovery following hospital discharge [7,8].

Globally, one-third of children between 6 and 23 months of age experience moderate to severe food insecurity, and only one-third consume an adequately diverse diet [9]. Most children with moderate or severe wasting live in resource-limited settings where household food insecurity and inadequate dietary diversity are critical drivers of poor nutritional outcomes [10,11]. Current guidelines for the management of children with wasting focus on nutritional supplementation and improved breastfeeding [12], despite the fact that many wasted children live in food-secure households. Dietary diversity may be important for these children, as high-nutrient-value diets such as legumes, nuts, and animal-derived foods, including milk, eggs, and meat, play a vital role in increasing the muscle mass of children and hastening weight gain [13,14,15]. It is plausible that children from food-secure households and/or those with adequate dietary diversity may experience slower rates of nutritional recovery than children whose primary cause of wasting was an insufficient caloric intake, as current interventions may not effectively address the underlying etiology of wasting in these children [8,16,17,18].

In this analysis, we evaluated the roles of reported household food insecurity, dietary diversity, and consumption of specific food groups at diagnosis of wasting during admission to hospital on the rate of postdischarge nutritional recovery among 6- to 23-month-old children living in resource-limited settings.

## 2. Methods

### 2.1. Study Design

We conducted a secondary analysis of the Childhood Acute Illness Network (CHAIN) Cohort, a multicountry prospective study of acutely ill children admitted to hospitals [19]. CHAIN recruited participants from nine hospitals across six countries; Bangladesh (icddr,b Dhaka Hospital, and Matlab Hospital), Burkina Faso (Banfora Regional Referral Hospital), Kenya (Kilifi County Hospital, Mbagathi Sub-County Hospital, and Migori County Referral Hospital), Malawi (Queen Elizabeth Central Hospital), Pakistan (Civil Hospital Karachi), and Uganda (Mulago National Referral Hospital) between 2016 and 2019. Hospitalized children were treated according to each country’s national recommendations, including for ‘severe acute malnutrition’, in hospitals and referred to a therapeutic feeding clinic at the hospital or near the child’s home [20]. Nutritional treatment of ‘moderate acute malnutrition’ was not included in the national guidelines, although all children at the icddr,b hospital received some supplementary food.

The CHAIN methods were published elsewhere [20]. In brief, CHAIN enrolled 2- to 23-month-old children hospitalized for acute illness and followed them for six months after hospital discharge. CHAIN’s exclusion criteria included not being able to tolerate oral feeds before the onset of the current acute illness, having an underlying illness requiring surgery within six months, having a diagnosis of a chromosomal abnormality, admission due to trauma or for surgery, requiring immediate cardiopulmonary resuscitation at admission, and a caregiver planning to move outside of the hospital catchment area within six months. Participants were allocated into three strata, deliberately over-representing moderate and severe wasting or kwashiorkor.

For this analysis, we included only children 6–23 months of age with wasting (MUAC < 12.5 cm) or kwashiorkor at the time of hospital discharge to evaluate factors associated with nutritional recovery postdischarge from the hospital. The weight-for-height Z-score was not included in the definition of wasting as diarrhea was highly prevalent in the cohort, and dehydration at admission would lead to substantial misclassification. As per the CHAIN strata for children aged 6 months and above, we defined severe wasting or kwashiorkor (SWK) as MUAC < 11.5 cm or nutritional edema and moderate wasting (MW) as MUAC 11.5–12.5 cm.

### 2.2. Outcome

The participants’ nutritional status was monitored during follow-up visits at 45, 90, and 180 days after hospital discharge. Participants underwent a thorough clinical examination at enrollment, at discharge from the hospital, and at each follow-up visit. Anthropometric measurements, including head circumference, MUAC, weight, and length, were taken by two independent study staff. The primary outcome of interest was time to recovery from wasting and its incidence. Recovery was defined as a MUAC > 12.5 cm without edema [4] at any follow-up visit.

### 2.3. Exposures

Exposures of interest included food insecurity, dietary diversity, and specific food types being consumed prior to hospital admission. Caregivers were interviewed to provide demographic and social information, including household food insecurity and children’s dietary diversity [21]. Food insecurity was measured using the household Food Insecurity Experience Scale (FIES) [22]. The FIES has eight yes/no questions that are answered with a score between 0 and 8. This score is further classified into food security (a score of 0 or having none of the food insecurity-related circumstances), mild food insecurity (a score of 1 to 3), moderate food insecurity (a score of 4 to 6), and severe food insecurity (a score of 7 to 8).

A standardized food frequency questionnaire was used to assess children’s dietary diversity, and data on specific food groups children consumed were gathered. Dietary diversity was evaluated using eight food groups: breastfeeding, milk and milk products, meats, legumes and nuts, roots and tubers, fruits and vegetables, grains, and eggs. Adequate dietary diversity was defined as having ≥2 food groups plus breast milk for children aged 6–10 months and ≥4 food groups plus breast milk for children older than ten months [23].

### 2.4. Statistical Analysis

Descriptive statistics were used to summarize children and caregivers’ baseline clinical and sociodemographic variables. Mean and standard deviations (SD) for continuous variables, counts, and percentages for categorical variables are presented. A chi-square test was used to analyze the relationship between dietary diversity and food insecurity. Participants were followed starting from hospital discharge until the first occurrence of recovery, death, loss to follow up, or end of the study. Participants’ outcome status was assessed based on the available MUAC and edema measurements. Participants were censored at the time of death or the date of the last follow-up visit. We added a one-month grace period after the 180th day; therefore, any follow-up completed beyond the 210th day was, administratively, censored. A Kaplan–Meier survival curve and cumulative incidence were used to describe the recovery probabilities. The log-rank test was used to test differences in the survival distribution of categories of selected variables.

A Cox proportional hazards model was fitted to estimate the association of food insecurity, dietary diversity, and specific food groups with time to recovery. We calculated crude hazard ratio (cHR) with 95% confidence interval. In the initial model, we estimated the association of food insecurity and dietary diversity with time to recovery. Then, a univariable model for each specific food group and a multivariable model comprising all specific food groups and confounders were fitted. The potential confounders in adjusted HR (aHR) models considered were study site, child age (continuous months), child sex, child HIV status (categorized as positive and negative), duration of hospitalization in days, MUAC (cm) at discharge, discharge to nutritional program (none, supplemental, or therapeutic), parental educational status (none, primary, and secondary or above), maternal/caregiver depression, the season of enrollment, residence (urban vs. rural), and household wealth status. Maternal or caregiver depression status was measured using the Patient Health Questionnaire-9 (PHQ9) tool, which is classified into three categories: mild, moderate, and severe. To account for seasonality, we considered the month of enrollment coupled with each month coded as rainy and not rainy for each site. Household wealth quintiles were calculated using principal component analysis (PCA), as previously published [24]. We evaluated the interaction of dietary diversity with food insecurity and their interaction (dietary diversity and food insecurity) with baseline nutritional status (SWK vs. MW) by likelihood ratio tests comparing models with and without the interaction terms. We evaluated interactions between age and dietary diversity, food insecurity, and individual food groups. We also stratified the effect of specific food groups on recovery by food insecurity status. We checked the proportional hazards assumption graphically and statistically using the Schoenfeld residual test. For variables that did not fulfill the proportional hazard assumption, we fitted a separate model containing an interaction with time to address the nonproportionality.

Finally, we conducted a sensitivity analysis to account for participants who did not have a follow-up MUAC measurement because of death, loss to follow-up, or missed measurements. We used two approaches: in the primary analysis, all participants were treated as censored at their last recorded study visit; in the second approach, these participants were excluded at baseline (a complete case analysis). Finally, we fitted a mixed-effects Cox proportional hazards model to understand if a random effect for a study site yielded different results. All analyses were performed using R version 4.0 (R core team (2021), Vienna, Austria).

### 2.5. Ethics

The ethical approval of the parent study was obtained from the Oxford Tropical Research Ethics Committee and ethics committees of all participating institutions [20]. Written consent was obtained from caregivers of all study participants for current and future use of study data and samples.

## 3. Results

### 3.1. Baseline Characteristics

A total of 3101 hospitalized participants were enrolled in the CHAIN cohort, 1286 (41%) of whom were six months of age or older and were wasted or had kwashiorkor at hospital admission. (Figure 1). Following discharge from the hospital, approximately half 587 (46%)) of participants had SWK, including 37/587 (6%) with kwashiorkor, and 699 (54%) had MW. The mean age of participants was 13 months (SD 5 months), with about half (627, 49%) aged between 6 and 11 months (61). Most participants (826, 64%) came from households reported to be food-insecure, including 170 (13%) with severe food insecurity (Table 1).

Approximately half of the participants (664, 52%) reported insufficient dietary diversity. The most frequently consumed food groups were grains (85%), breast milk (65%), and dairy products (59%). Meats (42%) and eggs (29%) were consumed the least. Children with MW reported lower levels of food insecurity (62% vs. 67%, *p* = 0.01) and higher dietary diversity (57% vs. 40%, *p* < 0.001) than children with SWK (Figure 2; Figure 3).

### 3.2. Association of Food Insecurity and Dietary Diversity

Participants reporting less food insecurity also reported greater dietary diversity (*p* < 0.001). Among participants from food-secure households, 274/460 (60%) reported sufficient dietary diversity compared with 141/349 (40%) and 58/170 (34%) children from moderate and severe food-insecure households, respectively (Appendix A).

### 3.3. Recovery from Wasting

During follow-up after hospital discharge, 825 (64%) participants recovered from wasting, 226 (27%) within 45 days, 301 (36%) between the 45th and 90th day, and 298 (36%) after the 90th day (Figure 4 and Appendix A). Of the 461 (36%) who did not recover during follow-up, 91 (20%) died and 22 (5%) withdrew or were lost to follow-up. The overall incidence rate of recovery was 18/100 child-months (825/4610, Table 2). Participants with MW experienced faster recovery, 25/100 person-months (95% CI: 23, 27), than those with SWK, 12/100 person-months (95% CI: 10, 13, *p* < 0.001). Age had a statistically significant association with recovery. Children older than 12 months had a 36% higher recovery rate than children who were younger than 12 months (cHR: 1.36 [1.19, 1.56], *p* < 0.001). However, age had no interaction with food insecurity, dietary diversity, or any of the food groups (Appendix A).

### 3.4. Association of Food Insecurity and Dietary Diversity with Recovery from Wasting

We observed no association between reported dietary diversity at baseline and the time to recovery (aHR 1.07 [0.91, 1.25]) *p* = 0.519). Food insecurity did not fulfill the proportional hazard assumption, and we therefore fitted a separate model containing the interaction of time with food insecurity. We also separately evaluated differences in recovery rate until the 90th day of follow-up (where the survival curves for food insecurity levels crossed) and after the 90th day of follow-up. However, the estimates were similar across these analyses (Appendix A, Appendix A).

Those with severe and moderate food insecurity had a 14% and 19% higher recovery rate (aHR 1.14 [0.88, 1.48] and 1.17 [0.96, 1.43]), respectively, than food-secure participants. Participants who reported insufficient dietary diversity had a slightly higher recovery rate than those with sufficient dietary diversity (aHR 1.07 [0.91, 1.25], Figure 5). However, these results were not statistically significant.

### 3.5. Association of Specific Food Groups with Recovery from Wasting

After adjusting for age, sex, HIV status, nutritional status at discharge, and household wealth, participants who reported eating legumes and nuts prior to admission had a 21% higher rate of recovery (aHR 1.21 [1.01,1.44]) than those who did not. Participants who had consumed fruits and vegetables had a 22% lower rate of recovery (aHR 0.78 [0.65,0.94]) than those who had not. Breastfeed children had a 16% lower recovery rate than non-breastfeeding children (aHR 0.84 [0.71, 0.99]). Consumption of dairy products (aHR 1.13 [0.96, 1.34]) and flesh foods (aHR 1.11 [0.93, 1.33]) were not associated with rate of recovery (Figure 6). Results from the sensitivity analyses, including the mixed-effects Cox regression model accounting for a random effect by the study site, were very similar (Appendix A, Appendix A).

## 4. Discussion

We sought to determine the association of reported food insecurity and dietary diversity measured prior to the diagnosis of wasting with recovery from wasting among 6–23-month-old children discharged from the hospital following an acute illness in resource-limited settings. Among this group of medically complicated children with wasting, prior dietary diversity was not associated with the recovery rate. We observed a trend toward a higher recovery rate among participants with insufficient dietary diversity, even after accounting for food insecurity and adjusting for age, site, and other anticipated confounders. In addition, we found that food insecurity was not associated with rates of recovery. It is likely that diet was improved through the provision of therapeutic feeds provided through nutrition programs, and nutritional counseling, but this management did not appear to be substantially more effective among children whose wasting was associated with dietary risk factors compared with children with other etiologies [25,26].

The consumption of legumes and nuts was associated with a 21% higher recovery rate in this study. Legumes and nuts are rich in protein, and the provision of protein-rich animal and plant products is recommended by the community management of malnutrition guidelines [13]. In resource-limited settings where animal source- proteins are less available [27], plant sources of protein may be valuable in meeting nutritional requirements. Interestingly, children who reported consuming fruits and vegetables had a 22% lower rate of recovery from wasting. In this study, the proportion of children who consumed fruits and vegetables was higher in study sites with the lowest rate of recovery (Appendix A; Appendix A). This result may be due to residual confounding by site or misreporting.

Breastfeeding is essential to maintaining healthy growth and development and should be complemented by other nutritious foods after the sixth month of life [28,29,30,31]. Breastfeeding for an extended period in resource-limited settings is important for preventing infection and improving overall children’s health [32]. In children over six months of age, we found that breastfeeding was associated with a 16% reduction in the rate of recovery, independent of age. Breastfeeding remains a critical intervention for the management of wasting, and the most important role of breastfeeding in wasted children may be to prevent medical complications rather than expedite weight gain. Additionally, it is important to note that this study included children aged 6 to 23 months of age, many of whom may had been weaning over the follow-up period (Appendix A, Appendix A). During weaning, children are vulnerable to poor weight gain, and the observed association may be attributable to ongoing efforts to wean the breastfed children in comparison with children who were already weaned at the time of hospital admission [33].

Children from households experiencing moderate and severe food insecurity may have had a modestly faster recovery rate than children from food-secure households. The management of wasting addresses food insecurity through the provision of supplemental calories and nutrients. Children with other reasons for malnutrition, including those with chronic underlying medical conditions such as HIV, sickle cell or congenital abnormalities, or those with difficulties feeding, may derive less benefit from nutritional supplementation compared with those who are suffering from poor food security or lack of dietary diversity (Figure 4). The majority of children in this study with severe wasting and medical complications came from food-secure or mildly food-insecure households. This also suggests that other underlying issues may be important contributors to malnutrition among medically complicated children with wasting [34,35]. Therefore, a holistic approach to treating wasting is superior to one exclusively focused on nutritional intervention.

This study has several strengths. We used data from a multicenter prospective cohort study that captured detailed dietary data in a harmonized and systematic manner across multiple sites. The study also has some limitations that need to be considered. The data collected in this study classified food groups from the current WHO classification by not differentiating vitamin-A-rich fruits and vegetables. This may limit the generalizability of our finding to other research studies. The measures we used to quantify food insecurity and dietary diversity assessed the children’s prehospital household environment and might not reflect their state during follow-up. However, social circumstances are unlikely to dramatically change before and after hospitalization. In addition, average calory intake, frequency of consumption of therapeutic foods, or changes in diet due to nutritional advice were not documented, which may have led to residual confounding and nondifferential misclassification. These influences may have also attenuated the role of dietary diversity and specific food groups on recovery rates. Additionally, the use of MUAC to assess wasting and recovery may have also led to misclassification, as MUAC alone does not capture all anthropometric measures of recovery. Finally, the follow-up time points in the study might not reflect the actual time when recovery occurred, which could have caused interval censoring, resulting in an underestimation of the rate of recovery.

## 5. Conclusions

Among wasted children discharged from hospital and supported with nutritional management, household dietary diversity and food insecurity, as assessed prior to the diagnosis of wasting, had minimal or no association with recovery. Modest differences in the recovery rates were observed among children consuming protein-rich plant sources prior to admission to the hospital. These data suggest that a child’s diet prior to becoming unwell does not appear to alter recovery from wasting. These findings suggest that children recovering from medically complicated wasting may require additional services beyond the provision of nutritional interventions to achieve optimal recovery. Novel interventions targeting factors other than food insecurity and dietary diversity, including those that focus on the complex social, medical, and nutritional issues contributing to malnutrition in this high-risk population, are urgently needed.

## Figures and Tables

**Figure 1 nutrients-14-03481-f001:**
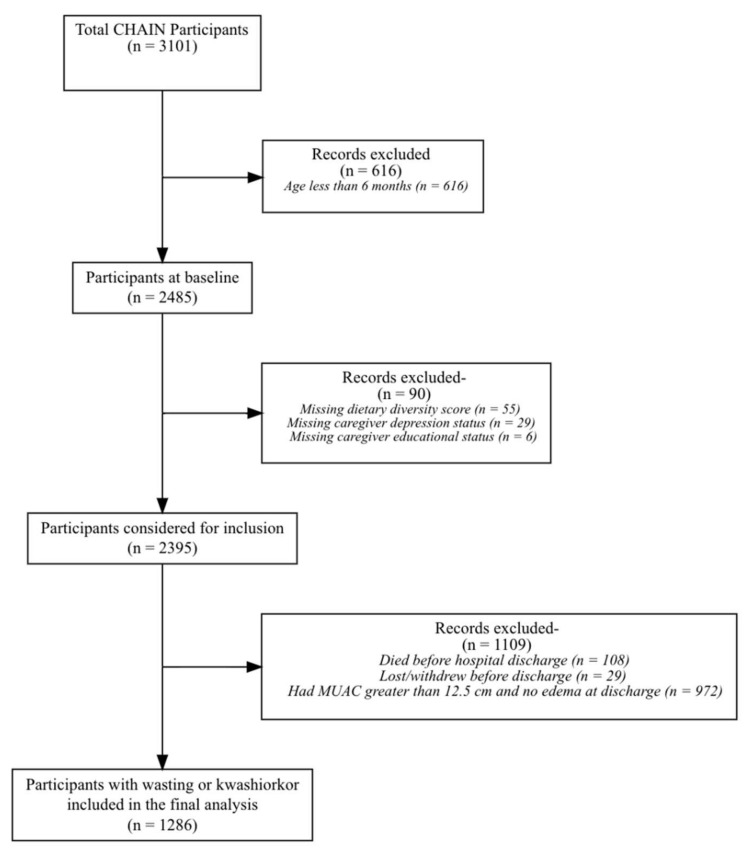
Flow chart for study participant inclusion.

**Figure 2 nutrients-14-03481-f002:**
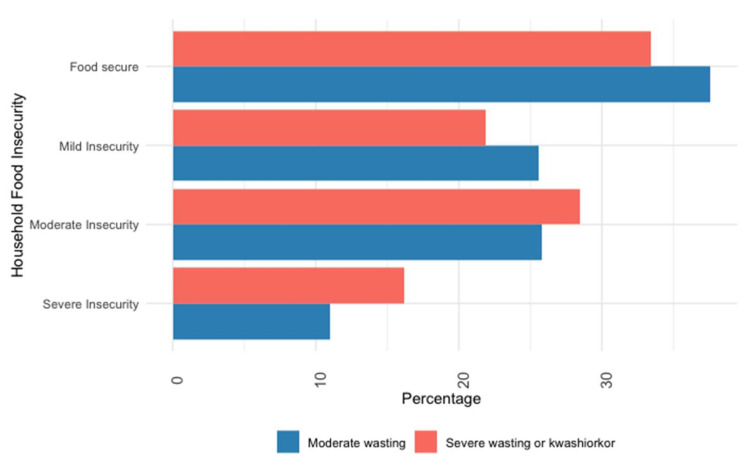
Distribution of food insecurity by nutritional status.

**Figure 3 nutrients-14-03481-f003:**
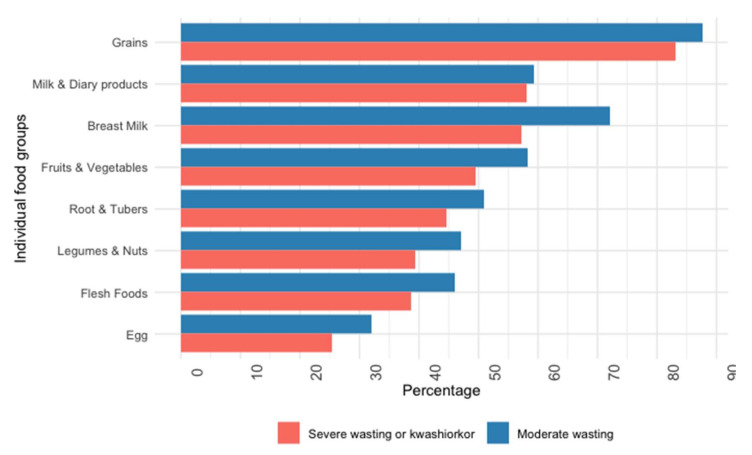
The proportion of specific food group consumption by nutritional status.

**Figure 4 nutrients-14-03481-f004:**
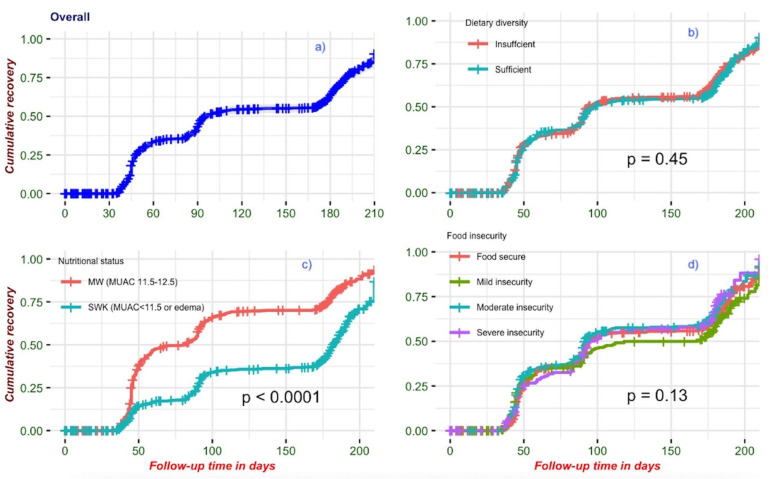
Kaplan–Meier curves for the cumulative rate of recovery: (**a**) overall cumulative recovery; (**b**) cumulative recovery by dietary diversity; (**c**) cumulative recovery by baseline nutritional status; (**d**) cumulative recovery by food insecurity status.

**Figure 5 nutrients-14-03481-f005:**
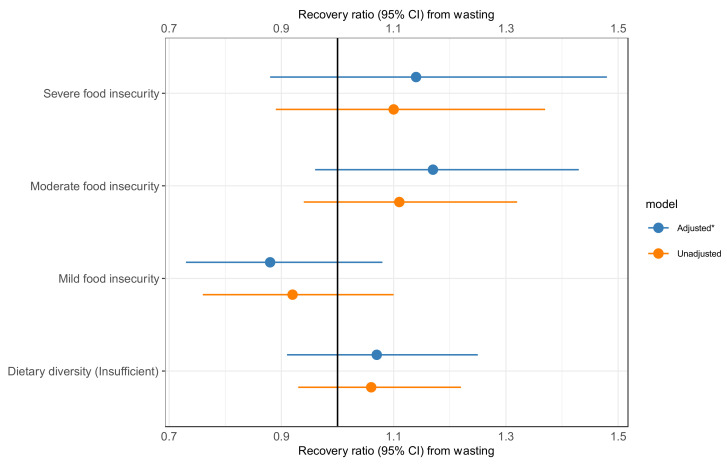
Association of food insecurity and dietary diversity with recovery from wasting. *Adjusted for child age, child sex, child HIV status, child baseline nutritional status, discharge to nutritional program, parental educational status, maternal/caregiver depression, the season of enrollment, residence, study setting, duration of hospitalization, and wealth status.

**Figure 6 nutrients-14-03481-f006:**
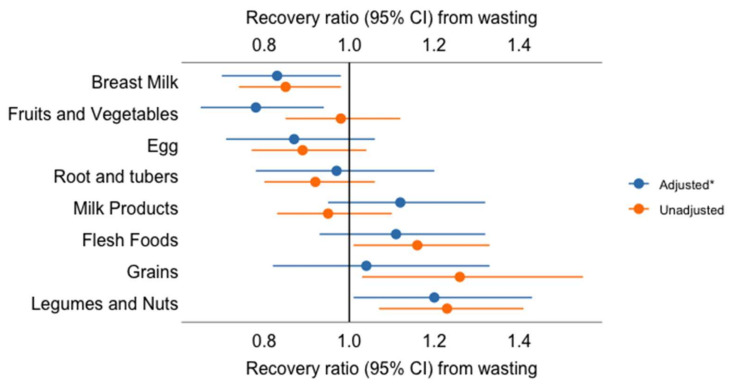
Association of specific food groups, food insecurity, and dietary diversity with recovery from wasting. ***** Adjusted for child age, child sex, child HIV status, child baseline nutritional status, discharge to nutritional program, parental educational status, maternal/caregiver depression, the season of enrollment, residence, study setting, duration of hospitalization, and wealth status.

**Table 1 nutrients-14-03481-t001:** Baseline children, caregiver, and household characteristics, and relationship with nutritional status at the time of hospital discharge.

Characteristic	MW ^a^ *n* = 6991	SKW ^b^ *n* = 5871	Overall *n* = 1286
**Age**			
6–11 months	330 (47%)	297 (51%)	627 (49%)
12–17 months	250 (36%)	186 (32%)	436 (34%)
18–23 months	119 (17%)	104 (18%)	223 (17%)
**Sex**			
Female	318 (45%)	286 (49%)	604 (47%)
Male	381 (55%)	301 (51%)	682 (53%)
**MUAC** ^ **c** ^	12.01 (0.32)	10.60 (0.89)	11.37 (0.95)
**Food security status**			
Food-secure	264 (38%)	196 (33%)	460 (36%)
Mild food insecurity	180 (26%)	127 (22%)	307 (24%)
Moderate food insecurity	180 (26%)	169 (29%)	349 (27%)
Severe food insecurity	75 (11%)	95 (16%)	170 (13%)
**Sufficient Dietary diversity**	394 (56%)	228 (39%)	622 (48%)
**Kwashiorkor at admission**	66 (9.4%)	133 (23%)	199 (15%)
**Kwashiorkor at discharge**	0 (0%)	37 (6.3%)	37 (2.9%)
**Stunting ***			
Severe stunting	185 (26%)	316 (54%)	501 (39%)
Moderate stunting	215 (31%)	156 (27%)	371 (29%)
No stunting	299 (43%)	112 (19%)	411 (32%)
Unknown	0	3	3
**Breastfeeding**			
Exclusive	472 (68%)	308 (52%)	780 (61%)
No	200 (29%)	258 (44%)	458 (36%)
Partial	27 (3.9%)	21 (3.6%)	48 (3.7%)
**HIV status**			
Negative	676 (97%)	543 (93%)	1219 (95%)
Positive	23 (3.3%)	44 (7.5%)	67 (5.2%)
**Primary caregiver educational status**			
None	196 (28%)	180 (31%)	376 (29%)
Primary	284 (41%)	260 (44%)	544 (42%)
Secondary and above	217 (31%)	145 (25%)	362 (28%)
Unknown	2	2	4
**Caregiver depression status (PHQ9)**			
None/minimal	231 (33%)	183 (32%)	414 (33%)
Mild	294 (42%)	234 (40%)	528 (41%)
Moderate	129 (19%)	105 (18%)	234 (18%)
Severe	40 (5.8%)	57 (9.8%)	97 (7.6%)
Unknown	5	8	13
**Wealth quintiles**			
Poorest	127 (18%)	118 (20%)	245 (19%)
Second	134 (19%)	118 (20%)	252 (20%)
Middle	142 (20%)	127 (22%)	269 (21%)
Fourth	157 (22%)	123 (21%)	280 (22%)
Least poor	139 (20%)	101 (17%)	240 (19%)
**Urban residence**			
Rural	357 (51%)	302 (51%)	659 (51%)
Urban	342 (49%)	285 (49%)	627 (49%)
**Country ****			
Bangladesh	200 (29%)	150 (26%)	350 (27%)
Burkina Faso	102 (15%)	94 (16%)	196 (15%)
Kenya	144 (21%)	137 (23%)	281 (22%)
Malawi	47 (6.7%)	45 (7.7%)	92 (7.2%)
Pakistan	71 (10%)	72 (12%)	143 (11%)
Uganda	135 (19%)	89 (15%)	224 (17%)

^a^: MW: moderate wasting; ^b^: SWK: severe wasting and kwashiorkor; ^c^: MUAC: mid-upper arm circumference. * Stunting was measured based on the height-for-age Z-score (<−3 severe, −3 to −2 moderate, and >−2 no stunting. ** Population density of >5000/km^2^.

**Table 2 nutrients-14-03481-t002:** Rate of recovery by food group.

Characteristics	N	Person Time (Months)	Recovered (*n*)	Recovery Rate (95% CI) (Per 100 Person-Months)	The Proportion of Recovered (%)
Overall	1286	4610	825	18(17, 19)	64
**Food groups**					
Grains	1099	3898	717	18(17, 20)	65
Breast Milk	828	3027	515	17(16, 19)	62
Milk and Dairy products	754	2722	491	18(16, 20)	65
Fruits and Vegetables	693	2489	453	18(17, 20)	65
Root and Tubers	617	2257	398	18(16, 19)	65
Legumes and Nuts	557	1931	383	20(18, 22)	69
Meats	545	1938	375	19(17, 21)	69
Egg	371	1385	237	17(15, 19)	64

## Data Availability

Data are publicly available at https://dataverse.harvard.edu/dataverse/chain (accessed on 3 February 2022).

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
