# Peer review of "The Role of Food Insecurity and Dietary Diversity on Recovery from Wasting among Hospitalized Children Aged 6–23 Months in Sub-Saharan Africa and South Asia"

_nutrients, 2022, doi:10.3390/nu14173481_

Round 1

Reviewer 1 Report

This secondary analysis of the Childhood Acute Illness Network cohort shows interesting results that could impact the follow up intervention on wasting children after discharge. The recovery period can benefit from additional services beyond nutritional supplementation interventions in order to achieve optimal recovery.

Major improvements needed

- If authors have access to the employment status, this data should be included in the potential confounders, since, as the analyzed educational status, is a strong determinant of income.

-Authors decided to exclude weight-for-height zscore in the definition of wasting to avoid dehydration cause by diarrhea at admission. They rely on MUAC WHO classification (and edema). Even this is a very useful methodology to access wasting in the community, authors should point possible limitations (Current MUAC Cut-Offs to Screen for Acute Malnutrition Need to Be Adapted to Gender and Age: The Example of Cambodia. Fiorentino M, et al.  DOI: 10.1371/journal.pone.0146442; Routinely MUAC screening for severe acute malnutrition should consider the gender and age group bias in the Ethiopian non-emergency context. Tessema M, DOI: 10.1371/journal.pone.0230502)

To access dietary diversity, the WHO 2017 determine 8 groups of foods. The 8 groups analyzed by authors are not the same as those determined by WHO (data provided in figure 2). The group grains and the group roots and tubers are considered just one group by WHO (grains, roots and tubers). This means that the group Vitamin-A rich fruits and vegetables was not analyzed by authors. The food group analysis should be done according to WHO reference or in not possible should be mention in text and discussed.

Minor adjustments:

Abstrat - “…on providing therapeutic and supplementary foods…” – the supplementary foods are included in the therapeutic, so the sentence needs reformulation;

Author analyze time to recovery from wasting during in the 6 months post-hospital discharge. The sentence “We evaluated if household food insecurity, dietary diversity, and the consumption of specific food groups prior to diagnosis of wasting are determinants of the duration of wasting treatment among hospitalized children.” suggests the evaluation was during hospital staying. Please rewrite sentence to be clearer;

Page 4, in results - authors present data on Stunting, but this was not part of methodology. Please add to methodology the description of all the variables analyzed in this a secondary analysis;

Table 1: Please provide in full the abbreviations used.

Author Response

Response: Thank you very much for this review – it was really helpful. Please find out response below.

This secondary analysis of the Childhood Acute Illness Network cohort shows interesting results that could impact the follow up intervention on wasting children after discharge. The recovery period can benefit from additional services beyond nutritional supplementation interventions in order to achieve optimal recovery. 

Major improvements needed 

- If authors have access to the employment status, this data should be included in the potential confounders, since, as the analyzed educational status, is a strong determinant of income. 

Response: Thank you very much for this valuable input. We originally did not include occupation in an effort to minimize over adjustment – as we assumed that this variable would be colinear with both education and wealth. However, we appreciate the comment and we have re-analyzed the data adding occupation as a potential confounder. The estimates remained did not change with the inclusion of occupation.

-Authors decided to exclude weight-for-height zscore in the definition of wasting to avoid dehydration cause by diarrhea at admission. They rely on MUAC WHO classification (and edema). Even this is a very useful methodology to access wasting in the community, authors should point possible limitations (Current MUAC Cut-Offs to Screen for Acute Malnutrition Need to Be Adapted to Gender and Age: The Example of Cambodia. Fiorentino M, et al.  DOI: 10.1371/journal.pone.0146442; Routinely MUAC screening for severe acute malnutrition should consider the gender and age group bias in the Ethiopian non-emergency context. Tessema M, DOI: 10.1371/journal.pone.0230502)

Response: Thanks for your comment and sharing the resources. We have highlighted the possible misclassification that could have been caused by only using MUAC in the limitations section.

To access dietary diversity, the WHO 2017 determine 8 groups of foods. The 8 groups analyzed by authors are not the same as those determined by WHO (data provided in figure 2). The group grains and the group roots and tubers are considered just one group by WHO (grains, roots and tubers). This means that the group Vitamin-A rich fruits and vegetables was not analyzed by authors. The food group analysis should be done according to WHO reference or in not possible should be mention in text and discussed. 

Response: Thank you for this very important point. We are aware that WHO 2017 divides food groups slightly differently than we have done in this analysis. Unfortunately, data were not collected in the parent trial in a way that allows us to mirror the WHO classification. We agree with the reviewer that this should be discussed in the manuscript and have added text to limitations sections to that effect.   

Minor adjustments:

Abstract - “…on providing therapeutic and supplementary foods…” – the supplementary foods are included in the therapeutic, so the sentence needs reformulation;

Response: Thank you for the comments – we have corrected this sentence and deleted the term “supplementary”

Author analyze time to recovery from wasting during in the 6 months post-hospital discharge. The sentence “We evaluated if household food insecurity, dietary diversity, and the consumption of specific food groups prior to diagnosis of wasting are determinants of the duration of wasting treatment among hospitalized children.” suggests the evaluation was during hospital staying. Please rewrite sentence to be clearer;

Response: Thank you for the comments – we have rewritten the statement for clarity.

Page 4, in results - authors present data on Stunting, but this was not part of methodology. Please add to methodology the description of all the variables analyzed in this a secondary analysis.

Response: Thank you for this comment. We wish to clarify that the stunting data presented is only shown as it relates to describing the study population and it’s not part of the outcomes of the study. We agree that readers will want to confirm our definition of stunting matched international norms and we have included the classification of stunting as a footnote in the results section (Table 1).

Table 1: Please provide in full the abbreviations used.

Response: Thank you, we have now provided these as footnotes.

Thank you again - Kirk Tickell & Adino Tsegaye

Reviewer 2 Report

Overall the structure and contents of the manuscrit fit with the aim. However I would suggest to add information regarding the characteristics of food insecurity and dietary diversity in different countries participating in the studies. Also, undergoing interventions of international organizations would be interesting to be mentioned.

Author Response

Thank you for this supportive review and  feedback. Please find our response below:

Overall the structure and contents of the manuscrit fit with the aim. However I would suggest to add information regarding the characteristics of food insecurity and dietary diversity in different countries participating in the studies. Also, undergoing interventions of international organizations would be interesting to be mentioned.

Response: Thank you so much for raising this. We have now highlighted current data on food security and dietary diversity in the study countries in the background section. In both settings we have note that national CMAM and IMAM guideline were followed, and these guidelines capture the majority interventions delivered to these children. We agree that there may have been NGOs delivering additional community level interventions. Unfortunately,  we do not have a mechanism  to retrospectively ascertain what NGOs were delivering nutrition sensitive interventions across these two relatively large geographies (Karachi city/Sindh province, and Migori County).